# Assembling Spheroids of Rat Primary Neurons Using a Stress-Free 3D Culture System

**DOI:** 10.3390/ijms241713506

**Published:** 2023-08-31

**Authors:** Meaghan E. Harley-Troxell, Madhu Dhar

**Affiliations:** Tissue Engineering and Regenerative Medicine, Large Animal Clinical Sciences, College of Veterinary Medicine, University of Tennessee, Knoxville, TN 37996, USA; mharley4@vols.utk.edu

**Keywords:** 3D culture, neurospheres, primary neural cells, nerve tissue engineering

## Abstract

Neural injuries disrupt the normal functions of the nervous system, whose complexities limit current treatment options. Because of their enhanced therapeutic effects, neurospheres have the potential to advance the field of regenerative medicine and neural tissue engineering. Methodological steps can pose challenges for implementing neurosphere assemblies; for example, conventional static cultures hinder yield and throughput, while the presence of the necrotic core, time-consuming methodology, and high variability can slow their progression to clinical application. Here we demonstrate the optimization of primary neural cell-derived neurospheres, developed using a high-throughput, stress-free, 3D bioreactor. This process provides a necessary baseline for future studies that could develop co-cultured assemblies of stem cells combined with endothelial cells, and/or biomaterials and nanomaterials for clinical therapeutic use. Neurosphere size and neurite spreading were evaluated under various conditions using Image J software. Primary neural cells obtained from the hippocampi of three-day-old rat pups, when incubated for 24 h in a reactor coated with 2% Pluronic and seeded on Poly-D-Lysine-coated plates establish neurospheres suitable for therapeutic use within five days. Most notably, neurospheres maintained high cell viability of ≥84% and expressed the neural marker MAP2, neural marker β-Tubulin III, and glial marker GFAP at all time points when evaluated over seven days. Establishing these factors reduces the variability in developing neurospheres, while increasing the ease and output of the culture process and maintaining viable cellular constructs.

## 1. Introduction

Neural injuries, caused by trauma and disease that disturbs the normal role of the peripheral nerves, spinal cord, and brain, affect millions of people globally; current treatments are ineffective at consistently restoring full function to the impacted tissue [1,2,3,4]. Advanced methods are needed to produce improved cellular therapeutics in order to meet the standard of or improve the efficacy of existing treatments. The use of 3D culture systems to develop spheroids has contributed to the field of regenerative medicine and neural tissue engineering by expanding opportunities to study processes of neuron development and repair. A spheroid is a loosely bound, cellular aggregate forming into a sphere-like cluster, that can further develop individually and in network formations [5,6]. More specifically, neurospheres are spheroids composed of primary neural cells (PNCs) [7,8,9,10]. Neurospheres, as a cell therapy, can contribute to axon growth, myelination, angiogenesis, immunoregulation of the microenvironment, and proliferation and differentiation of the endogenous cells. Each contribution addresses a flaw in the nervous system’s capacity to repair post-injury [10,11,12,13,14]. Through additional contacts and release of neurotrophic factors into the microenvironment, neurospheres exhibit increased cell–cell interactions as compared to cells in suspension [5,6]. For these reasons, neurospheres adopt an *in vitro* environment that mimics, at a fundamental level, the natural *in vivo* conditions of the nervous system, bridging the gap in translational medicine. This will accelerate the timeframe to progress the science from benchside to clinic for both animal and human studies by improving the complexity of the culture to match that of the established internal structures. Understanding these intricacies will facilitate the advancement of neural therapeutics.

Numerous techniques have been used to develop spheroids for a wide range of tissue engineering applications, including the hanging drop technique, spinner flasks, and encapsulation in hydrogels [6,15]. These practices use gravitational forces, microfluidics, and mimicked extracellular matrix (ECM) behavior to induce spontaneous assembly of cells, maximizing cell–cell cohesion, while simultaneously minimizing interactions with outside sources. However, challenges of 3D cultures remain unaddressed. Firstly, current methods are often time-consuming and costly with low yields and large variation in neurosphere size [6]. Additionally, many cultures use stem cells, such as mesenchymal stem cells (MSCs) and induced pluripotent stem cells (iPSCs). These cell types require longer time periods to differentiate into neurospheres and have exhibited inconsistency in their cell proliferation, differentiation, and gene expression [6,13,14]. Incorporating PNCs as an alternative accelerates the timeline of differentiation to hours while decreasing the variability in individual cell behavior [16,17].

While previous studies have shown neurospheres to improve and accelerate healing after neural injury as compared to cells in suspension, one of the most significant challenges remains the presence of a necrotic core [10,18,19]. The lack of vascularization hinders gas and chemical nutrient exchange and the removal of cellular waste, resulting in cellular death at the center of the neurosphere. Necrotic cores scale with neurosphere size, a product of the seeded cell population or time in culture. In stationary cultures, these conditions produce a hypoxic environment that impacts neurosphere quality. Because of this challenge, culturing multi-cellular spheroids through a stress-free method that continuously washes the cellular assembly while promoting aggregation can minimize tissue stress and offer an approach to further develop co-cultured cellular populations toward tissue and organoid constructs [6].

Here we employ a new commercial, high-throughput 3D culture system to establish neurospheres derived from PNCs. A single 3D bioreactor provides a stress-free environment to produce a 10-fold increase in the number of neurospheres compared to the conventional 96-well U-bottom plate. This *in vitro* method alleviates the challenges of a co-culture process and produces neurospheres with a more efficient and cost-effective approach [9]. In this work, we optimize this technique to develop neurospheres for therapeutic use. With our long-term goal to generate a regenerative medicine strategy to effectively treat neural injuries, we aim to establish *in vitro* culture methods that will reduce the variability and enhance the effectiveness of neurospheres. We hypothesize that the establishment of these tissue engineering variables provides us with optimal neurospheres that could be implanted *in vivo* to improve functional recovery post-injury.

## 2. Results and Discussion

### 2.1. Three-Day Old Pups Provided the Largest Number of Cells for the ClinoStar

Hippocampi of one-to-four-day-old rat pups were used to collect the PNCs for this experiment. As each reactor ideally holds between two and four million cells for a range of cell types, we first identified the time that could obtain the most cells for the *in vitro* cell culture with a starting population of three million cells. After 49 dissections across four timepoints, day three resulted in the highest number of cells, reaching up to 17.1 million cells collected from five animals (Figure 1). Day one hippocampi are small and can be challenging to dissect, while day four tissue sustains greater damage from advanced neuronal differentiation and tissue development. With this information, the number of animals may be increased or decreased depending on the number of cells that result from the isolation process. The number of reactors needed for an experiment result from the number of parameters tested in the reactor conditions, balanced by the cellular yield. Where substrate treatments are the test variable for the end objective of the study, fewer reactors can be used. From our experience, one reactor yields a high number of PNC-derived neurospheres adequate for most studies. It may be noted that some variation occurs with person-to-person experience in both these and future results.

### 2.2. Pluronic ClinoReactor Coating and PDL Plate Coating Result in the Largest Neurosphere Size

We evaluated the effects of changing surface chemistry and its impact on neurosphere size. Neurosphere size was determined as the neurosphere diameter and the neurite spreading surface area [12]. A large range in neurosphere size has previously been reported using conventional 3D culture methods. Average diameters ranging from less than 100 µm to greater than 500 µm is common, but considerations of necrotic core development determine the lower end of the range to be preferable [12,15]. Measurements of neurite spreading area is less commonly documented, with one study finding a range of 390–610 µm^2^, and another finding a range of 1416–6252 µm^2^. However, increased neurite spreading means greater cell–cell communication for enhanced neurosphere effects [6,20].

Differences in neurosphere size were first analyzed across the ClinoReactor coatings: Pluronic and Biofloat. For comparison, uncoated ClinoReactors were used as a negative control. Neurospheres developed in ClinoReactors coated with Pluronic resulted in a mean neurosphere diameter of 109 μm, a significantly larger size than those coated in Biofloat (*p*-value: 0.0042), which produced a mean neurosphere diameter of 90 µm. Yet the use of either coating created neurospheres with significantly greater diameters than those in an uncoated ClinoReactor (*p*-values: 0.0001 and 0.0001), which had a mean diameter of 47 µm (Figure 2A). Similarly, neurospheres established in Pluronic-coated ClinoReactors resulted in the largest neurite spreading surface area, as compared to both Biofloat and uncoated ClinoReactors (*p*-values: 0.0003 and 0.0001, respectively), with a mean surface area of 233,164 μm^2^. Biofloat had a mean surface area of 155,845 μm^2^, which was also a significantly larger mean surface area than the uncoated reactor, with 39,968 μm^2^ (*p*-value: 0.0001) (Figure 2B).

We utilized ClinoReactor coatings to repel the cell attachment to the reactor surface. This allows the cell-to-cell interactions to develop the neurospheres because it minimizes the interactions between the cells and the reactor surface. The Biofloat coating solution was developed to form spheroids by preventing non-specific protein binding that is both reproducible and improves upon the plastic and glass plates that are often used for spheroid development, whereas Pluronic is a well-known co-polymer composed of poly(ethylene oxide) (PEO) and poly(propylene oxide) (PPO) that creates a hydrophobic surface to prevent cell adhesion [21,22]. Both coatings were easy to use and time-efficient, though Pluronic is more cost-efficient. Though both coatings were effective in their goals, underscoring the importance of a ClinoReactor coating, our results demonstrated Pluronic-coated ClinoReactors to produce neurospheres that were both significantly larger in neurosphere diameter and neurite spreading surface area. This may suggest that Pluronic was more effective at reducing non-specific interactions when developing neurospheres.

We then investigated the differences in neurosphere size across the plate coatings: Laminin + PDL, PDL, PO, and TCPS. Laminin + PDL and PDL surfaces had the largest mean neurosphere diameters, with 95 μm and 88 μm, respectively, exhibiting no significant differences between them (*p*-value: 0.3593). However, both Laminin + PDL and PDL produced significantly larger neurospheres relative to both PO, with a mean diameter of 66 μm (*p*-values: 0.0030 and 0.0001, respectively), and TCPS, with a mean diameter of 59 μm (*p*-values: 0.0044 and 0.0001, respectively). There were no significant differences between PO and TCPS coatings (*p*-value: 0.3137) (Figure 2C). In contrast, Laminin + PDL coating had the largest neurite spreading surface area as compared to PDL, PO, and TCPS coatings, with a mean surface area of 112,826 μm^2^ (*p*-values: 0.0006, 0.0003, and 0.0001, respectively). PO coatings resulted in a mean neurite spreading surface area of 46,177 μm^2^, which was not significantly different from coatings PDL (*p*-value: 0.1199) and TCPS (*p*-value: 0.0507). However, the PDL-coated plates resulted in neurospheres whose neurite spreading surface area was significantly greater than TCPS (*p*-value: 0.0001), with a mean surface area of 66,884 μm^2^ as compared to 32,370 μm^2^ (Figure 2D).

In contrast to the repellant ClinoReactor coatings, the plate coatings were used after the neurospheres were removed from the ClinoStar incubator to provide additional functional groups and proteins for improved cellular support and neurosphere growth *in vitro*. Previous studies have shown PDL to be a highly favorable cell adhesion substrate for neural cell attachment, while laminin has proven to enhance neural cell growth and axon guidance [23,24,25]. We combined these proteins to promote neurosphere development *in vitro*. Our data found PDL to enhance neurosphere growth when compared to the standard tissue culture plates, and adding laminin did not significantly promote an additional change in neurosphere size. However, the addition of laminin did enhance neurite growth as compared to PDL alone. Meanwhile, PO is a synthetic coating, similar to PDL, proven to enhance neural cell growth [26]. Our data showed that PO did not enhance neurosphere or neurite growth to the extent that PDL did. Yet PDL and PO both use amines as their functional groups for better attachment [12]. With the knowledge that additional ECM support enhances attachment to aid in neurite outgrowth, future studies may look at how different ECM molecules or additional cellular support would change the differentiation and composition of neurospheres.

### 2.3. Twenty-Four-Hour Incubation Time Results in Larger Neurite Outgrowth

In addition to evaluating the effects of surface chemistry, we also analyzed the differences in neurosphere size across two different incubation times: 24 h and 48 h. No significant differences were determined between the neurosphere diameter at 24 h and at 48 h, with a mean value of 89 μm and 98 μm, respectively (*p*-value: 0.1132) (Figure 3A). However, evaluation of the neurite spreading surface area showed significant differences between the two incubation times, with the mean value 155,845 μm^2^ at 24 h showing significantly larger neurite surface area as compared to the mean value 101,582 μm^2^ at 48 h (*p*-value: 0.0001) (Figure 3B).

With all coating variables remaining consistent (Pluronic, PDL), our results demonstrated that time in the ClinoStar incubator did not significantly alter the neurosphere diameter. When we evaluated neurite spreading surface area, the neurospheres incubated for an additional 24 h did not allow for increased neurite spreading. Instead, incubating the neurospheres for an additional 24 h restricted the neurite outgrowth. This limitation may be due to a number of *in vitro* limitations, including limited availability of growth factors and other nutrients overtime, that may be provided by fresh supplemented media [27]. Because cell–cell contacts have substantial effects on cell development, future studies may look at how cell number and neurosphere size increase the need for media changes as compared to 2D cultures.

### 2.4. Supplemented Media Is Required for Appropriate Neurosphere Size and Neural Differentiation

Though it is well established that media supplements are necessary for density dependent growth of healthy cell cultures and spheroid development, we assessed how cell–cell contacts influence neurosphere outcomes through a media supplement (B-27 and GlutaMAX™) deprivation study using the stress-free “continuous-bathing” system of the ClinoStar incubator [28,29,30]. For samples without B-27 and GlutaMAX supplements, we observed high cell viability (≥80%) maintained throughout the cultures up to five days *in vitro*. However, the neurosphere size was significantly decreased, with very little neurite growth across the coating conditions and incubation times that were most successful in previous experiments. Another consequence of the supplement deprivation is the expression of microglial marker Iba1 (Figure 4). Microglia are known to initiate an inflammatory response, migrating to an injured area [31]. The presence of microglia, using the marker Iba1, is often seen as a sign of inflammation, damage, and degeneration in neural culture, which is used to study traumatic brain injury and neurodegenerative diseases [32]. Overall, the neurospheres were strongly influenced by the conditions of their media in the first 24 h of incubation. Though further research is needed, the removal of supplements that typically benefit neural cultures do not appear to be compensated by improved cell–cell contact and affect the fate of the PNCs *in vitro*. Future studies may look into how additional supplements, such as neurotrophic factors, influence cell composition for neurosphere development.

### 2.5. Neurospheres Maintain Viability and Retain Neural and Glial Cell Types through 5 Days In Vitro

To complement the size analysis and confirm that the increased size is not plagued with a simultaneous increase in cellular apoptosis, we assessed the viability of the neurospheres over time, at days one, three, five, and seven *in vitro*. Mean percent viability ± standard deviation was 84.11% ±15.37 on day one, 89.31% ±14.03 on day three, 99.32% ±1.65 on day five, and 100.00% ±0.00 on day seven (Figure 5A,B). The neurospheres maintained good viability (≥80%) across the seven-day period. When filtering variables for reactor coatings, plate coatings, and incubation time, the viability did not significantly change. Though seven days should be sufficient time to develop neurospheres of satisfactory cell quality and behavior for therapeutic treatments, future studies may look at longer time frames.

Through immunofluorescence, we visualized the expression of neural and glial cell types by the neurospheres at 1, 3, and 5 days *in vitro*. Due to neurosphere size and fast process of neural differentiation that would accelerate the timeline for future therapeutic treatments, a five-day time period was chosen as a starting point for cellular integration and co-culture studies *in vitro* and *in vivo* [33]. The mature neural cells were visualized through the expression of MAP-2 and B-Tubulin III antibodies, while the glial cells were visualized through the expression of GFAP (Figure 5C–E). Variables in coating conditions (Pluronic, PDL) and incubation time (24 h) were kept consistent throughout the experiments. All three antibodies were expressed consistently across the three timepoints. Our data confirm a quick, efficient, and consistent cell behavior from these neurospheres that could benefit further nerve growth repair studies.

## 3. Materials and Methods

### 3.1. Biochemicals, Chemicals, and Disposables

All biochemicals, cell culture supplements, and disposable tissue culture supplies were purchased from Thermo Fisher Scientific (Waltham, MA, USA) unless otherwise noted.

### 3.2. Animals

Wistar Kyoto rat breeder pairs were obtained from Japanese sources Ueta et al. and bred against wild-type rats obtained from Charles River Laboratories. All procedures were conducted in accordance with PHS guidelines for the humane treatment of animals under approved protocols established through the University of Tennessee’s Institutional Animal Care and Use Committee (IACUC) (Knoxville, TN, USA) (IACUC# 2508-1219).

### 3.3. Bioreactor

Bioreactors were prepared prior to cell collection as per manufacturer recommendations (CelVivo, Odense, Denmark, celvivo.com). Briefly, a syringe with 20 ½ gauge needle was used to dispense 25 mL sterile water (CelVivo, Odense, Denmark) into the hydration port of the ClinoReactor^®^ (CelVivo) under sterile conditions. The ClinoReactor^®^ (CelVivo, Odense, Denmark) was left for 4 h at room temperature or placed in a 4 °C fridge overnight with the hydration port and two air vents covered to prevent contamination. Using the front port, the culture chamber was washed twice with sterile PBS. Five milliliters of either BIOFLOAT™ FLEX coating solution (faCellitate, Mannheim, Germany) or 2% Pluronic F-127 solution was used to coat the reactor. A ClinoReactor^®^ without any coating was used as a negative control. The culture chamber was washed twice with PBS. The ClinoReactor^®^ was filled to the top of the port, removing air bubbles, with 10 mL neurobasal media. The ClinoReactor^®^ was placed in the ClinoStar^®^ at 37 °C, 5% CO_2_, 15 rpm, for one to two hours (Figure 6).

### 3.4. Primary Neural Cell Collection and Expansion

Pups (one-four days old) were rapidly decapitated, brains were extracted, and then hippocampi regions were dissected as described previously. The papain enzyme was prepared to produce 20 units per mL hibernate, pH 7.0–7.4 [34,35]. The hippocampi were pooled into 5 mL enzyme solution, and placed in a water bath for 30 min, agitating the tissue approximately every 10 min. The enzyme solution was aspirated, and the tissue was suspended in 2 mL Hibernate with no phenol red. Tissue was dissociated using a fire-polished Pasteur pipette. The dissociated tissue was then transferred to a 15 mL tube. The suspension, dissociation, and transfer were repeated a second time. The tube was centrifuged at 0.4 rcm for five minutes. Afterwards, the solution was aspirated, the pellet was dislodged, and then cells were resuspended in 5 mL neurobasal media. Cells were counted using a 10 µL cell solution with 10 µL trypan blue [25,36]. A master solution was created, seeding three million cells in 10 mL neurobasal media per bioreactor. ClinoReactor^®^ bioreactors were filled to the top of the port, removing air bubbles. Placed bioreactors in the ClinoStar^®^ at 37 °C, 5% CO_2_, 3 rpm, for 24 to 48 h.

### 3.5. Media

Hibernate-A and Neurobasal-A media, with and without phenol red, were supplemented with 2% B-27, 1% Penicillin and Streptomycin, and 0.25% GlutaMAX™. All experiments were performed with supplemented media, except those documented as non-supplemented. Non-supplemented media was Hibernate-A and Neurobasal-A medium only [25,36]. Media was warmed in a water bath to 37 °C, unless otherwise noted.

### 3.6. Neurosphere Cultures

Neurospheres were verified after 24 h by viewing the ClinoReactor^®^ under a light microscope. Under sterile conditions, the neurospheres were seeded onto 10% Laminin (Ln) + Poly-D-Lysine (PDL), PDL, and Poly-L-Ornithine (PO)-coated plates. Five hundred microliters of the coating were used to condition each well of a tissue-cultured polystyrene (TCPS) 12-well plate for 1 h. TCPS with no coating was used as the negative control. Each coated well was washed with PBS and used. One ClinoReactor^®^ containing 3,000,000 cells driven to aggregation was divided among multiple wells at approximately 250,000 cells per well. Additional supplemented neurobasal media was added to each well to bring the total volume to 1 mL in each well. The cells were placed in the 37 °C incubator until the specific time points described below.

### 3.7. Viability Staining

Neurospheres were stained with 5 mM Calcein AM + 1.5 mM propidium iodide (PI) solution to test for viability as described previously [37,38]. For staining, neurospheres were incubated at 37 °C for 15 min. Fluorescent images were taken immediately after incubation using Leica DMi8 microscope. Neurospheres were evaluated at days one, three, five, and seven post-seeding.

### 3.8. Immunofluorescence

Immunofluorescent staining was performed as described previously [33,37]. Cells were fixed with 4% paraformaldehyde (PFA) for 10 min at room temperature at specific time points. Cells were permeabilized with 0.1% Triton for 10 min at room temperature. Cells were blocked with 1% Universal Blocking Reagent (BioGenex, Fremont, CA, USA) for 30 min at room temperature, and incubated with specific antibodies overnight at 4 °C. The primary antibodies used were 1:200 fibrous-actin (FA) (Invitrogen, Waltham, MA, USA), 1:500 microtubule-associated protein 2 (MAP2) (Biolegend, San Diego, CA, USA), 1:500 beta-III tubulin (TuJ1) (Biolegend, San Diego, CA, USA), and 1:500 glial fibrillary acidic protein (GFAP) (Invitrogen, Waltham, MA, USA), ionized calcium-binding adaptor protein 1 (Iba1) (Fujifilm, Tokyo, Japan). Corresponding secondary antibodies were used the next day: 1:500 goat anti-chicken (Invitrogen, Waltham, MA, USA), 1:500 rabbit anti-mouse (Invitrogen, Waltham, MA, USA), and 1:500 donkey anti-rabbit (Invitrogen, Waltham, MA, USA). Secondary antibodies were added and kept at room temperature for 30 min. After washing, cells were mounted with Prolong™ gold antifade reagent with DAPI (Invitrogen, Waltham, MA, USA). After 24 h, cells were imaged at 10× magnification using Leica DMi8 microscope, obtaining at least five images from different areas in each well.

### 3.9. Image J Analyses

Fiji/Image J software was used to obtain data for statistical analysis from phase contrast microscope images [39]. A 10× magnification image of a hemacytometer was used to set the scale for all image analyses. Neurosphere diameter was measured using the straight-line tool drawn across each neurosphere of phase contrast images. “Measure” was selected and length documented. Neurite spreading surface area was measured using the freehand selection tool drawn around the periphery of the neurosphere and neurites of phase contrast images. “Measure” was selected and area documented [12]. Quantitative immunofluorescence data were analyzed for percent area. Images were opened, split color channels (red or green), enhanced contrast (0.3), adjusted the threshold (100, 254), converted to mask, watershed, and analyzed for particles (0.05, infinity).

### 3.10. Statistics

All data were exported to Microsoft Excel worksheets. Data were analyzed for mean and standard deviation values, controlled for conditions of time incubated, time *in vitro*, reactor coatings, and plate coatings. A Shapiro–Wilk test of normality was performed using GraphPad Prism (version 10). Data were not normally distributed, hence, the two-tailed Mann–Whitney test was used to compare each coating variable in Figure 2 and each time variable in Figure 3. *p* ≤ 0.05 was considered significant.

## 4. Conclusions

The use of the ClinoStar incubator and ClinoReactor has provided a high-throughput method that produces a large number of neurospheres in a short period of time. As a result, this method reduces the reagents needed, improves efficiency, and reduces variation. The stress-free environment provides a technique that alleviates the current challenges of neurosphere development. By studying reactor coatings, plate coatings, incubation times, and media on neurosphere diameter, neurite spreading, cell viability, and neural differentiation, we have determined that using our methodology, the optimal conditions to develop neurospheres for therapeutic treatments will require Pluronic reactor coatings, PDL + Laminin plate coatings, 24 h incubator time, and five days *in vitro*. Though these parameters are recommended, there is flexibility in the conditions that would retain good cultures. In addition, PNCs obtained from three-day-old pups are a preferred cell choice to develop neurospheres due to the accelerated timeline of neural and glial differentiation and low variability in cell behavior, which produces high yields for cultures [16,17].

By applying these baseline variables established for efficient neurospheres, and with our long-term goal to generate a regenerative medicine strategy to effectively treat neural injuries, there are now many directions that these neurospheres can be taken. Ongoing work in our lab includes the integration of neurospheres with other cell types, such as MSCs, that could provide an additional ECM, or endothelial cells, that could provide an additional blood supply to the growing neurospheres [40,41]. Future directions may also incorporate biomaterials and nanomaterials, such as graphene and its derivatives, which could provide additional mechanical and electrical properties to enhance the therapeutic effects of the culture [10,19]. With this advance, neurospheres have the potential to develop into an effective therapeutic treatment.

## Figures and Tables

**Figure 1 ijms-24-13506-f001:**
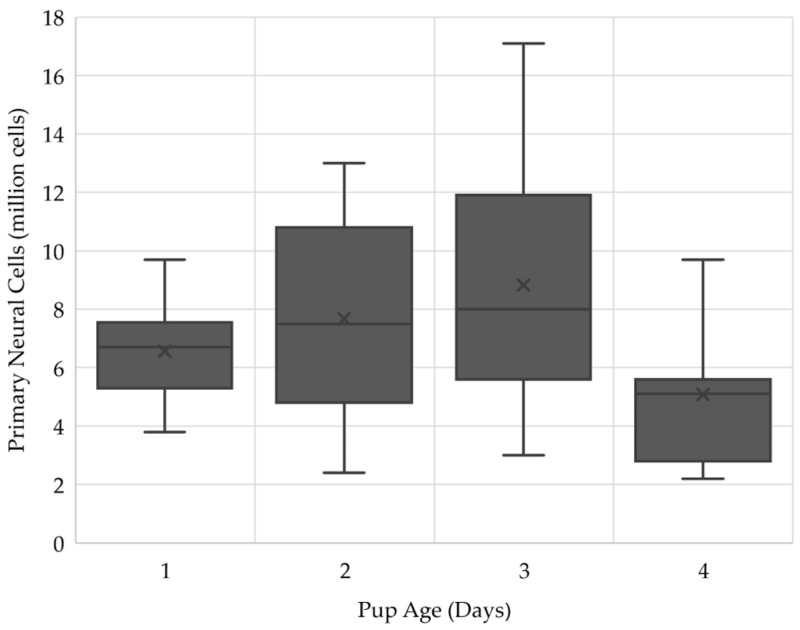
PNCs collected from the hippocampi of five rat pups at one- to four-days old. Three-day old pups provided the largest number of cells for the ClinoStar reaching up to 17.1 million cells.

**Figure 2 ijms-24-13506-f002:**
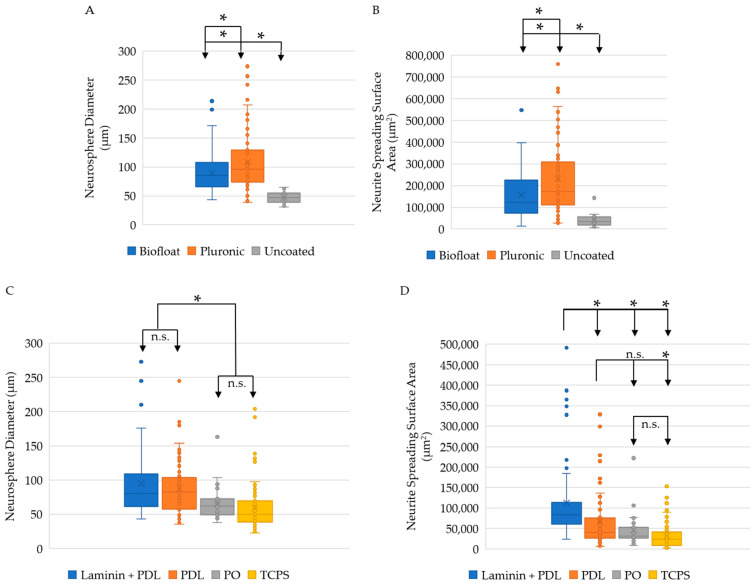
Neurosphere size is dependent on ClinoReactor and plate coatings. ClinoReactor coatings Pluronic and Biofloat, and plate coatings Laminin + Poly-D-Lysine (PDL), PDL, Poly-L-Ornithine (PO), and Tissue Culture Polystyrene (TCPS) result in changes to neurosphere diameter and neurite spreading surface area. (**A**) Pluronic coatings significantly increased neurosphere diameter (mean diameter = 109 µm) as compared to the Biofloat coatings (mean diameter = 90 µm) and uncoated ClinoReactors (mean diameter = 47 µm). (**B**) Pluronic coatings significantly increased neurite spreading surface area (mean surface area = 233,164 µm^2^) as compared to Biofloat coatings (mean surface area = 155,845 µm^2^) and uncoated ClinoReactors (mean surface area = 39,968 µm^2^). (**C**) Neurosphere diameter (μm) mean values are 95, 88, 66, and 59, respectively, with Laminin + PDL and PDL plate coatings presenting the significantly largest neurosphere size. (**D**) Neurite spreading surface area (μm^2^) mean values are 112,826, 66,884, 46,177, and 32,370, respectively, with Laminin + PDL plate coating presenting the significantly largest neurite spreading surface area. * *p* ≤ 0.05. n.s. denotes no significance.

**Figure 3 ijms-24-13506-f003:**
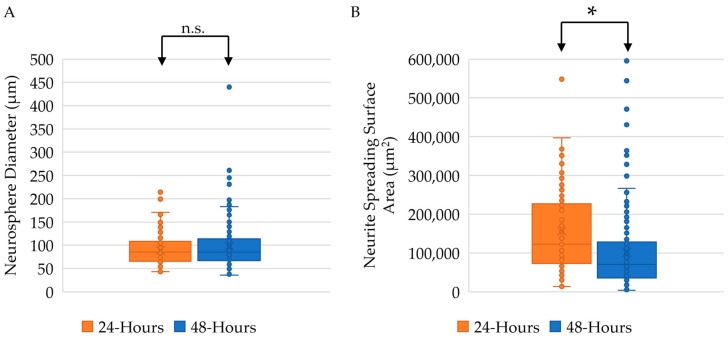
Neurosphere size is dependent on time in ClinoStar incubator (**A**). Neurosphere diameters compared at 24 h and 48 h incubation times, with mean values of 89 μm and 98 μm, respectively, showed no significant differences. (**B**) Neurite spreading surface area compared at 24 h and 48 h incubation times, with mean values of 155,845 μm^2^ and 101,582 μm^2^, respectively. Significant increase in neurite spreading seen at 24 h. * *p* ≤ 0.05. n.s. denotes no significance.

**Figure 4 ijms-24-13506-f004:**
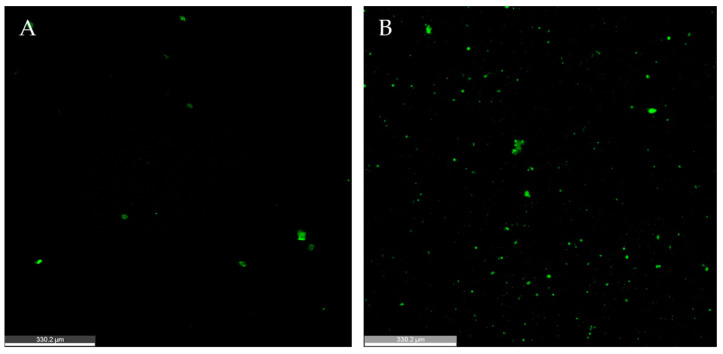
Density dependence of neurospheres cultured without media supplements. Media with B27 supplement, GlutaMAX showed (**A**) reduced neurosphere size, lacking neurites, maintained high cell viability in this Calcein image, and (**B**) expression of Iba1 exhibits presence of microglia. Scale bar = 330.2 µm.

**Figure 5 ijms-24-13506-f005:**
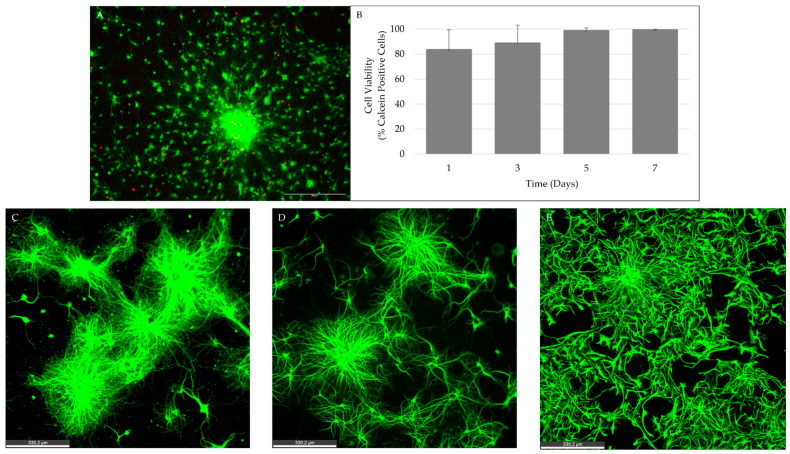
Neurosphere viability and neural and glial differentiation of neurospheres over 5–7 days *in vitro*. (**A**) Fluorescent images of Calcein (green, live cells) and Propidium Iodide (PI) (red, dead cells) for day 5 neurosphere. (**B**) Percent viability of neurospheres at days 1, 3, 5, and 7 *in vitro*. Neurosphere cultures maintained high viability (≥80%) across the 7-day period. (**C**) MAP2 and (**D**) B-Tubulin 3 are expressed by neural cells; (**E**) GFAP is expressed by glial cells in day 3 neurospheres. Scale Bar (**A**) = 300 µm; Scale bar (**C**–**E**) = 330.2 µm.

**Figure 6 ijms-24-13506-f006:**
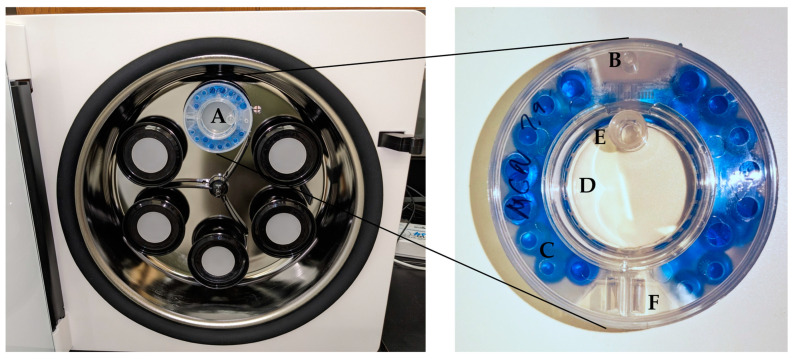
ClinoStar^®^ incubator and ClinoReactor^®^. (**A**) Holds up to 6 ClinoReactor^®^ bioreactors. (**B**) Is the port for the sterile water to hydrate the (**C**). Water beads that create a humidification system. The water beads are separated by a semi-permeable membrane from the (**D**). Growth culture system, which can be accessed from the (**E**). Front and (**F**). Top ports.

## Data Availability

The data that supports the findings of this study are available on request from the corresponding author, M.D.

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
