# Peer review of "Assembling Spheroids of Rat Primary Neurons Using a Stress-Free 3D Culture System"

_ijms, 2023, doi:10.3390/ijms241713506_

Round 1
Reviewer 1 Report
I have reviewed the article and found it can be accepted after major revision.
What is the potential of neurospheres in the field of regenerative medicine and neural tissue engineering?
What are the challenges associated with implementing neurosphere assemblies?
What is the methodology used to optimize primary neural cell-derived neurospheres?
How was neurosphere size and neurite spreading evaluated in the study?
What type of cells were used to develop neurospheres in the study?
What coating was used in the reactor and plates for the development of neurospheres?
How long does it take to establish neurospheres suitable for therapeutic use using the methodology described in the study?
Also refer to the following related article as
10.22034/JNA.2022.680836
I have reviewed the article and found it can be accepted after major revision.
What is the potential of neurospheres in the field of regenerative medicine and neural tissue engineering?
What are the challenges associated with implementing neurosphere assemblies?
What is the methodology used to optimize primary neural cell-derived neurospheres?
How was neurosphere size and neurite spreading evaluated in the study?
What type of cells were used to develop neurospheres in the study?
What coating was used in the reactor and plates for the development of neurospheres?
How long does it take to establish neurospheres suitable for therapeutic use using the methodology described in the study?
Also refer to the following related article as
10.22034/JNA.2022.680836
Reviewer 2 Report
Title
Assembling Spheroid of Rat Primary Neurons Using a Stress-Free 3D Culture System
Overall view
The new techniques are used to study growth conditions of primary nerves, The article has clear ideas and data and has made a credible control group, Research on new technologies should be encouraged.
Highlight
Use a variety of markers to prove the growth of nerve cells.
Inadequacy
In the experiment, the author shows the growth of nerve cells in the first week, Did the authors study the results over a longer period?
What is the difference in results between cell culture using the technique in this article and other techniques?
Decision
This manuscript had a clean result.
Accepted
